# Two to Tango? The Dance of Maternal Authority and Feeding Practices with Child Eating Behavior

**DOI:** 10.3390/ijerph18041650

**Published:** 2021-02-09

**Authors:** Ada H. Zohar, Lilac Lev-Ari, Rachel Bachner-Melman

**Affiliations:** 1Clinical Psychology Graduate Program, Ruppin Academic Center, Emek Hefer 40250, Israel; ldlevari@gmail.com (L.L.-A.); rachel.bachner@mail.huji.ac.il (R.B.-M.); 2Lior Zfaty Center for Suicide and Mental Pain Research, Emek Hefer 40250, Israel; 3School of Social Work, Hebrew University of Jerusalem, Jerusalem 91905, Israel

**Keywords:** overt-covert feeding styles, maternal authority style, maternal feeding practices, child eating behavior

## Abstract

The purpose of this study was to elucidate the relationship between maternal feeding practices and children’s eating problems. Mothers of 292 children aged 5.9 ± 1.1, 50% boys, reported online on parental authority, overt and covert control of the child’s food choices, child feeding practices, and their child’s problematic eating behavior. Structural equation modelling yielded a model with excellent indices of fit (χ^(2)^_(52)_ = 50.72, *p* = 0.56; normed fit index (NFI) = 0.94; root mean square error of approximation (RMSEA) = 0.001). The model showed that an authoritarian maternal authority style was associated with overt control, which was associated with maternal tendency to pressure children to eat and with maternal restriction of highly processed or calorie-rich snack foods. These, in turn, were positively associated with the child’s satiety response, food fussiness, and slow eating, and negatively with the child’s enjoyment of food. In contrast, a permissive maternal authority style was associated with covert control of the child’s eating, concern over the child being overweight, and the restriction of highly processed and calorie-rich snack foods, which were in turn positively associated with the child’s emotional overeating and the child’s food responsiveness. The model seems to tap into two distinct patterns of mother-child feeding and eating dynamics, apparently related to children with opposing appetitive tendencies.

## 1. Introduction

This study focuses on the connections between maternal authority styles, feeding practices and early childhood eating problems. Parenting style is the parental attitudes to rules and to disciplining their children [1], and this can result in individual differences in key child outcomes. Baumrind [2] described four parenting styles, based on two dimensions (responsiveness and demandingness), which have been widely used in research: authoritative, authoritarian, permissive and neglectful. The first three styles can be assessed by self-report and have been studied in the context of feeding and eating. Feeding practices and child eating behaviors interact with parenting style [3,4].

Authoritative parenting, a democratic style high in both responsiveness and demandingness that provides rules and a positive context, is the style associated with the development of the healthiest child feeding habits [4]. Authoritative mothers, more than other mothers, tend to provide the warmth, support and limits needed for children to internalize positive behaviors such as self-regulation of eating behaviors [5,6]. For example, authoritative parenting is associated with fruit and vegetable consumption during childhood [7], lower obesity levels [8] and healthy diets in adolescent children [9].

Parental authority styles are related to parental feeding practices. In a systematic review and meta-analysis, Collins et al. [10] combined seven studies and found that the authoritarian parenting style was associated with pressuring the child to eat and with restricting the child’s diet; the authoritative parenting style was related to parental monitoring of high-calorie foods, while permissive parenting style was not. In a later review and meta-analysis using a cluster analytic approach, van der Host and Sleddens [11] found that authoritative parents were more likely to monitor the child’s intake of high calorie food, to model healthy eating and promote it, while authoritarian parents tended to control their toddlers’ food intake for weight control, and to use food as a reward.

Parenting practices are more context-dependent and less trait-like than parenting styles and can differ when applied to different children in the same family [4]. Parents can influence their children’s eating behavior via their feeding practices, by efforts to restrict or monitor their child’s food supply and food intake or by pressuring him/her to eat [12]. Research has consistently shown that excessive parental control (pressuring and restricting) has negative consequences for child eating behaviors and tends to be counterproductive [4,13,14]. Maternal parenting style is a possible risk factor for disordered eating behaviors in adolescents [15]. Deliberate attempts to restrict the consumption of unhealthy foods, such as forbidding sweets, tend to spotlight the restricted foods, making them more tempting and encouraging children to eat in the absence of hunger [16]. Maternal pressure to eat via prompting, rewarding or coercion is associated not with increased food consumption, but with avoidance of the food that the child is pressured to eat [17]. The concept of parental feeding control has been extended to covert and overt control that parents use to encourage healthy eating and discourage unhealthy eating in their children [18]. Overt control strategies include restricting and monitoring food intake and are communicated explicitly and verbally, while covert control is exerted non-verbally and indirectly. Overt control is associated with pressuring the child to eat [19,20,21], particularly with children who are underweight; longitudinal research shows that these efforts do not change their underweight status over 3 years of longitudinal observation [19]. Covert control is associated with the parental practice of restricting the child’s calorie-rich snack foods [19,20,21], particularly with children who are overweight; again, longitudinally these efforts do not change the overweight status of the child [19].

Hubbs-Tait et al. [22] mapped parenting styles onto child feeding practices and found that feeding practices with young children predicted authoritative, authoritarian and permissive parenting styles. Authoritarian style was predicted by restriction, pressure to eat, and a lack of monitoring. Authoritative style was predicted by responsibility, monitoring, modeling and a lack of restriction, and permissive style was predicted by restriction and a lack of modeling. Links between child eating behavior and parental feeding practices have also been examined in research [23,24,25]. In a Brazilian cross-sectional study of maternal feeding practices and nutritional guidance of 5–9 year-olds [26], there were positive relationships between maternal feeding behavior and the modeling of healthy eating, restriction of the child’s screen time (phones, tablets, computers, etc.) and better child health outcomes.

Connections between maternal eating, maternal feeding, and child problem eating are of obvious importance, yet they are understudied. Viana et al. [27] studied a large sample of Portuguese mothers and children. The mother’s eating could be characterized as one of three distinct patterns: restrictive calorie-counting-eating, overeating, and neutral, i.e., not restricting and not overeating. The use of maternal feeding practices and child eating behavior problems differed between these three groups. Mothers with the neutral eating pattern restricted their children’s eating less and pressured them to eat more than the other groups and were less concerned that their child was overweight. Their children were reported to be slow, fussy eaters with low food responsiveness and high satiety. The preschool period is critical for the development of communication between parents and children about food and eating [4], and it presents a window of opportunity for effective interventions. It is therefore important to understand the interplay between parenting styles, parental feeding practices and child eating behaviors during early childhood.

Although maternal feeding and child eating have been studied, the interplay of these processes with parental authority has not been elucidated. The current study examines the concurrent dynamic of maternal feeding and perceived problematic child eating behaviors. The pattern of associations between maternal authority style, maternal feeding practices and child eating behaviors is observed in six-year-old children. Due to their cross-sectional nature, there is no way to disentangle child characteristics and behaviors from maternal authority and feeding styles. Our study is therefore descriptive and focuses on how children and mothers respond to each other, and which maternal and child behaviors are most closely related. We hypothesized that maternal child feeding practices would mediate the effect of maternal authority style on child eating behaviors.

## 2. Method

### 2.1. Protocol

This study was embedded within a longitudinal project. In 2012–2014, we recruited a large baseline sample of women with a child between the ages of 2 and 5, who were proficient enough in Hebrew to self-report on an extensive questionnaire [28]. In 2014–2016, the participants who had expressed consent to be contacted in the future and had provided email addresses at baseline were contacted and sent a link to the online Qualtrics© questionnaire [29]. Just over 50% of those approached responded, resulting in a self-selected sample of women volunteers, who were Hebrew speaking and internet-connected.

### 2.2. Participants

Participants were 292 Jewish Israeli mothers, aged 32.8 ± 4.8 years. Mothers were highly educated, with 16.3 ± 2.2 years of schooling, and almost all (97.2%) lived with a partner. Family income placed them mainly in average (30.5%) or higher than average (57.9%) socio-economic status. Their children (50% boys) were 5.9 ± 1.1 years of age and had 0–9 siblings; 19.2% were only children and 48% had one sibling. Over half (58.6%) of the children were firstborn.

### 2.3. Measures

Parental authority style was assessed by the Parental Authority Questionnaire (PAQ) [30,31]. This 30-item self-report scale is composed of three subscales that correspond to three of Baumrind’s [32] parental styles: permissive, authoritarian, and authoritative. Mothers responded to each item on a five-point scale from 1 (“strongly disagree”) to 5 (“strongly agree”). The PAQ in Hebrew has adequate discriminant and criterion related validity [33]. A sample item from the permissive subscale is “I feel that in a well-run household, children should be free to behave as they see fit to the same extent as parents”. An example from the authoritarian subscale is “When I tell my child what to do, I expect immediate and unquestioning obedience”, and a typical item from the authoritative subscale is “Whenever we establish a family policy, we discuss its rationale with the children”. The internal consistency of the three scales in this study was, respectively, α = 0.86, 0.83 and 0.90.

Child eating behavior was assessed by the Child Eating Behavior Questionnaire (CEBQ) [34], a 35-item questionnaire assessing potentially problematic eating styles in young children. Items are scored on a 5-point Likert-like scale ranging from 1 (never) to 5 (always) and load onto the following 8 subscales: Food Responsiveness, e.g., “My child’s always asking for food”; Emotional Overeating, e.g., “My child eats more when anxious”; Enjoyment of Food, e.g., “My child loves to eat”; Desire to Drink, e.g., “My child is always asking for a drink”; Satiety Responsiveness, e.g., “My child gets full up quickly”; Slowness in Eating, e.g., “My child takes more than 30 min to finish a meal”; Emotional Undereating, e.g., “My child eats less when s/he is upset”; and Food Fussiness, e.g., “My child decides that s/he doesn’t like a food even without tasting it”. The subscale scores are the mean of the responses on the scale-items. In this study, we used a validated Hebrew translation [35]. The subscales showed good reliability, with Cronbach alpha values ranging between 0.83 and 0.95.

Parental beliefs, attitudes, and practices regarding child feeding were measured using the 31-item Child Feeding Questionnaire (CFQ) [12]. Items are scored on a 5-point Likert-like scale ranging from 1 (never) to 5 (always) that load onto the following 7 subscales: Perceived Responsibility of the Mother for Feeding the Child, e.g., “When your child is at home, how often are you responsible for feeding her?”; Perceived Weight Status of Child throughout Development and until the Present; Perceived Weight of Mother from Birth until the Present—both were answered on a scale between “very thin” and “extremely overweight”; Parental Concerns about Child (Over)Weight, e.g., “How concerned are you about your child eating too much when you are not around her?”; Monitoring, i.e., keeping tabs on the high-fat, high-sugar high-salt or processed foods the child eats, e.g., “How much do you keep track of the high-fat foods that your child eats?”; Restriction—the extent to which parents restrict their child’s access to certain foods they deem unhealthy, e.g., “I have to be sure that my child does not eat too many sweets (candy, ice cream, cake or pastries)”; and Pressure to Eat—the extent to which parents exert pressure on their children to eat more food than they want, e.g., “If my child says “I’m not hungry”, I try to get her to eat anyway”. In this study, we used the Hebrew version, which has excellent construct reliability [13]. Subscales showed good reliability, with Cronbach alpha values ranging between 0.56 and 0.83.

*Maternal approach to child food choices* was assessed by Overt-Covert control (OC) [18]. This scale includes 5 items that assess overt control, or control that can be detected by the child, and 5 items that assess covert control, or control that cannot be detected by the child. The OC scale was translated for use in the current study by translation, back-translation, comparison and revision. A sample item of overt control is “how often are you firm about when your child should eat?” And a sample item of covert control is “How often do you avoid buying sweets and crisps and bringing them into the house?” The response scale has five values ranging from “never” to “always”. The OC has good convergent validity versus the CFQ [18]. In the current study, the scale reliabilities for overt and covert control were 0.62 and 0.84, respectively.

### 2.4. Procedure

The study was approved by the Ruppin Academic Center IRB (#2016-35 L/cp). This was Time 2 (T2) of a longitudinal study [36]; Time 1 (T1) had taken place two years previously. Participating mothers provided informed consent and reported via an online questionnaire delivered through Qualtrics [29].

### 2.5. Data Analysis

All variables were tested for normality of their distribution and were found to be adequate. Pearson correlations were used to assess the associations between variables. A structural equation model (SEM) was used to assess the mediating effect of maternal feeding practices on the association between parenting styles and child eating behaviors. We chose the following values for acceptance of the model: normed fit index (NFI) > 0.90 [37], root mean square error of approximation (RMSEA) < 0.08 [38] and standardized root mean residual (SRMR) < 0.08. SPSS 23.0 (IBM, Armonk, NY, USA) and AMOS 23.0 (IBM, Chicago, IL, USA) were used for statistical analyses.

## 3. Results

Pearson correlations between parental style, child feeding practices and child eating behaviors are reported in Table 1.

As can be seen from Table 1, whereas parental authority styles were not significantly associated with child eating behaviors, feeding practices and child eating behaviors were highly correlated.

A Structural Equation Model (SEM) was built to assess the mediating effect of child feeding practices on the association between parental authority style and child eating behaviors. We entered all three authority styles, maternal feeding practices and child eating behaviors into the model (see Figure 1). The model showed acceptable fit ($\chi_{\left(53\right)}^{2}$ = 53.26; *p* = 0.46; Comparative Fit Index (CFI) = 1.00; Tucker-Lewis Index (TLI) = 0.99; NFI = 0.94, RMSEA = 0.004; standardized root means square residual (RMR) = 0.04). No direct effects were observed between parenting styles and eating behaviors. Indirect effects were observed between the authoritarian parenting style and Food Enjoyment, Food Fussiness, Slow Eating, Satiety Responsiveness, Restrictive Child Feeding Practices and Pressure to Eat. Indirect effects were also observed between the permissive parenting style and Food Responsiveness, Food Enjoyment, Emotional Under- and Overeating, Restrictive Child Feeding Practices and Concern about Child (Over-)Weight.

It should be noted that several of the study variables entered into the AMOS analysis were excluded from the model presented in Figure 1: of the three maternal authority styles, the authoritative authority style did not associate meaningfully with the feeding practices and was therefore omitted. Other variables were also omitted for clarity because they were not strongly associated with the other variables. The excluded variables were four of the seven CFQ subscales: Perceived Responsibility of the Mother for Feeding the Child, Perceived Overweight Status of the Mother, Perceived Overweight Status of the Child, and Maternal Monitoring of Child Food Intake. Of the eight CEBQ subscales, Desire to Drink was dropped.

For the variables included in the model, there are two main paths of association. An authoritarian maternal style was associated with overt control of child food choices and with the feeding practices Pressure to Eat and Restriction. Pressure to Eat was associated with child appetitive behaviors; positively with Satiety Response, Food Fussiness and Slow Eating, and negatively with Food Enjoyment. A permissive maternal style was negatively associated with covert control of child food choices and positively with Concern about Child (Over-)Weight, which in turn was associated with elevated child appetitive behavior, i.e., Emotional Overeating and Food Responsiveness.

## 4. Discussion

In this study, we found an integrative cross-sectional picture that ties maternal authority style to maternal feeding practices, which in turn relate to problematic eating behaviors of six-year-old children. The results of this study are encapsulated in Figure 1, and results of interest include the variables dropped from the model as well as those that were included due to their strong paths to other variables.

Authoritative parenting style is not included in the model. Authoritative parenting has many advantages for child development and is prospectively protective against child obesity [39]. Thus, it is not surprising that it does not feature in a model focusing on problematic child eating behavior. This can be seen as further proof that authoritative parenting is helpful for the child’s development of healthy eating from early childhood through to adolescence [4,6,7,8,9]. This result is also consistent with a recent systematic review by Burnett et al. [11], who found that authoritative parenting was related to the healthiest food intake in toddlers.

In contrast, an authoritarian parenting style was the starting point of many strong paths. It connected with overt control over the child’s eating choices, i.e., laying down the law for what, when, and how the child eats [18]. It was also significantly connected to two maternal feeding practices: pressuring the child to eat more than (s)he wants, and restricting his/her intake of calorie-rich or highly processed food. Pressuring the child to eat was related in turn to non-appetitive child eating behaviors: food fussiness, slow eating, satiety response, and non-enjoyment of food. These two paths connecting maternal feeding practices with child appetitive tendencies have been found in other studies and in diverse cultures [19,20,21].

It is tempting, though unjustified, to look at the direction of the arrows denoting the paths in Figure 1 and interpret the non-appetitive behavior of the child as a response to the authoritarian, overtly proscriptive parenting approach and pressure on the child to eat. Parenting authority style is very closely related to parental characteristics that predate parenting, and thus may be independent of the particular child and his/her needs. In particular, an authoritarian maternal style is related to lower levels of the maternal self-regulatory character traits of cooperation and self-directedness, which are related to a more anxious and less warm and outgoing temperament, in turn related to insecure attachment [40].

However, the interpretation that authoritarian parenting is the root cause of the child’s under-eating and food fussiness is no doubt an overstatement. The long-term influence of maternal feeding practices on the child’s eating behavior, though detectable, is weak [36]. It may be, in part, a response to the child’s appetitive tendencies. Individuals retrospectively reporting higher maternal pressure to eat during their childhood tend to have lower body mass index (BMIs) and healthier eating attitudes [13] as adults than those reporting lower maternal pressure to eat during their childhood. Moreover, Alfonso et al. [19] concluded their careful longitudinal study of maternal feeding and child’s BMI with the following statement: “We found that parents both respond to and influence the child’s weight; thus, this child–parent interaction should be considered in future research”. Thus, the dynamic apparent in this study via maternal report when the children are six years of age may be a snapshot without long-term effects on the child’s eventual eating attitudes, arising among other causes from the child’s appetitive tendencies. Nevertheless, clinicians interviewing the parents of children with restrictive eating disorders in which this dynamic interplay is present tend to blame maternal feeding practices [41]. Furthermore, although these practices may not be helpful, they may well be responses to low child appetitive tendencies and/or result from maternal characteristics.

In contrast, a permissive maternal authority style was associated with covert control of the child’s eating; i.e., trying to control what, when, and how the child eats in ways that were not detectable to the child [18], such as not stocking certain foods or avoiding restaurants that serve food of which the parent disapproves. Covert control was associated with maternal concern over the child being overweight, as well as with maternal restriction of the child’s consumption of highly processed and calorie-rich snack foods [12], which were in turn positively associated with the child’s emotional overeating and the child’s positive enjoyment of food (food responsiveness). The developmental pathways for maternal authority style have been investigated in the past on a subset of the participants [40]. A permissive maternal authority style was found to be related to maternal characteristics that predated maternity, specifically to the character trait of cooperation and the temperament trait of novelty seeking, which stabilize in early adolescence. Novelty seeking is positively related to anxious attachment which is infantile in origin [42]. Again, as the permissive maternal style is deeply related to individual pre-maternal characteristics, it is tempting to interpret the pathways from maternal permissiveness to child overeating or high BMI as causal. Such connections have been observed in other studies. In a cross-sectional study of 904 girls and boys aged 8 to 13 and their mothers in Germany [43], maternal restriction and monitoring were related to child episodes of loss of control and other pathological eating symptoms only for children with high BMI. The authors stressed that causality could not be assigned, yet pointed out the ineffectiveness of maternal restricting and monitoring efforts in limiting the food intake of highly appetitive children. Adults with high BMIs recalled experiencing more maternal concern with their weight and more maternal restriction and monitoring of their calorie-rich, low-density-nutrition snack foods during their childhood than adults with lower BMIs [13]. The two processes of parental feeding and eating associated with opposing child appetitive tendencies were observed in a longitudinal study of children four to seven years of age [19]. In this study, Afonso et al. concluded that temporal relationships suggested the influence of the child’s BMI over the parents feeding practices rather than the reverse.

### Limitations

This study has several limitations that restrict the generalizability of the results. The participants were highly educated middle-class women, while research has shown that feeding and eating are greatly influenced by education and income [44]. Feeding practices were measured using the CFQ. Although this instrument is widely used, it focuses on a limited number of feeding practices with an emphasis on highly controlling feeding. The participants were all Israeli Jews, and there is good evidence that ethnicity influences maternal feeding [45]. The participants were community volunteers, and the children they reported on were apparently all typically developing children and may not generalize to clinical cases. The study relied on a single source of data: maternal reports. The dynamic of the feeding–eating interaction was therefore filtered through the mother’s perception and may have been different if viewed through objective eyes [46]. The design is cross-sectional, so results are correlational, reflecting a single measurement point in time. However, since this “snapshot” aspect of a life-long dynamic is the very focus of this paper, it is not necessarily a limitation.

## 5. Conclusions

We found that an authoritarian maternal authority style was associated with overt control, which was associated with maternal tendency to pressure children to eat and restriction of their calorie-rich foods. These maternal practices were associated with children’s fussy eating, emotional undereating, and satiety response. In contrast, a permissive maternal authority style was associated with maternal covert control of children’s eating, concern over children being overweight, and restriction of their calorie-rich foods, which were in turn associated with child emotional overeating and high food responsiveness. These results seem to reflect two distinct patterns of mother-child feeding and eating dynamics that may correspond to different child appetitive tendencies. It takes two to tango, and the current study shows the intense interaction between mother and child in the development of problem feeding and problem eating.

## Figures and Tables

**Figure 1 ijerph-18-01650-f001:**
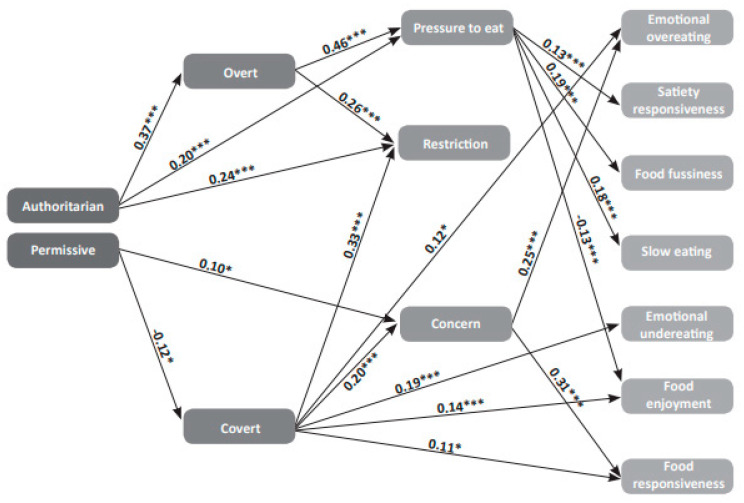
The mediating effect of child feeding practices on the relationship between maternal authority style and child eating behaviors. Note. * *p* < 0.05; *** *p* < 0.001. Rectangles denote calculated subscales.

**Table 1 ijerph-18-01650-t001:** Pearson correlations between parental style, child feeding practices and child eating behaviors.

	Permissive	Authoritarian	Overt	Covert	Concern	Restriction	Pressure
Emotional Overeating	0.06	0.03	0.02	−0.19 ***	0.30 ***	0.14 *	−0.02
Satiety Responsiveness	0.07	0.00	−0.01	−0.11	−0.16 **	−0.04	0.15 *
Food Fussiness	0.01	0.10	0.15 *	−0.06	−0.08	0.10	0.20 ***
Slow Eating	0.02	0.03	0.13 *	−0.06	0.00	0.02	0.18 **
Emotional Undereating	0.02	0.01	0.12 *	0.18 **	0.07	0.12 *	0.10
Food Enjoyment	−0.07	−0.06	0.02	0.20 ***	0.14 *	0.08	−0.14 *
Food Responsiveness	0.00	0.00	0.07	0.21 ***	0.37 ***	0.20 ***	−0.08

Note: Permissive, Authoritarian = Parental Authority Questionnaire; Overt; Covert = Overt-Covert control; Concern = Concern about Child (Over)Weight (CFQ, Child Feeding Questionnaire); Pressure = Pressure to Eat (CFQ).* *p* < 0.05; ** *p* < 0.01; *** *p* < 0.001.

## Data Availability

Data is available upon request from the corresponding author.

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
