# Peer review of "Two to Tango? The Dance of Maternal Authority and Feeding Practices with Child Eating Behavior"

_ijerph, 2021, doi:10.3390/ijerph18041650_

Round 1

Reviewer 1 Report

This was a very interesting article, one that would be of interest to those who work in this area of research or practice. The suggestions that are listed here are very minor. Overall, the research appears accurate and conducted well. Please see the suggestions listed below.

Introduction: The introduction provides good context for the study. There were some underdeveloped paragraphs (one sentence- e.g. lines 54-57; 64-67) and this could be improved by either elaborating on the content or finding a way to combine paragraphs. One addition this research provides is an assessment of overt AND covert feeding practices and the potential correlation to parenting styles and child eating behaviors. Additional information about covert feeding styles in the first mentioned 1 sentence paragraph could provide additional context for this interesting finding in the paper, while addressing the aforementioned structure issue. 

Lines 77-80 discuss the study design before even mentioning the purpose of the study. Suggest to mention this after text starting online 81 that describes the purpose of this study. 

Methods: Suggest to add additional text to 2.4 Data Analysis to describe the specific models that were created instead of describing them in the results section.  In your limitations, there were several factors that you mentioned in the limitations that could be associated with the results. Was it possible to include any of these in any preliminary analyses to determine whether these factors were likely to be associated with any of the analyses? If these were conducted, it might be worthwhile to indicate this briefly in the methods. 

Results: No additional suggestions here beyond the statement about the describing the model development in the methods section. 

Discussion: This section was well written and very interesting. 

Conclusions: Focus on the results from this study. Any similar results from other studies should be highlighted in the discussion section, in my opinion. 

Author Response

This was a very interesting article, one that would be of interest to those who work in this area of research or practice. The suggestions that are listed here are very minor. Overall, the research appears accurate and conducted well. Please see the suggestions listed below.

Introduction:

  1. The introduction provides good context for the study. There were some underdeveloped paragraphs (one sentence- e.g. lines 54-57; 64-67) and this could be improved by either elaborating on the content or finding a way to combine paragraphs.

This has been corrected.

  1. One addition this research provides is an assessment of overt AND covert feeding practices and the potential correlation to parenting styles and child eating behaviors. Additional information about covert feeding styles in the first mentioned 1 sentence paragraph could provide additional context for this interesting finding in the paper, while addressing the aforementioned structure issue. 

The description of overt and convert control in the context of child feeding has been extended, and some of the salient research pertaining to them as been reviewed in the Introduction.

  1. Lines 77-80 discuss the study design before even mentioning the purpose of the study. Suggest to mention this after text starting online 81 that describes the purpose of this study. 

This has been corrected.

Methods:

  1. Suggest adding additional text to 2.4 Data Analysis to describe the specific models that were created instead of describing them in the results section. 

This has been corrected.

  1. In your limitations, there were several factors that you mentioned in the limitations that could be associated with the results. Was it possible to include any of these in any preliminary analyses to determine whether these factors were likely to be associated with any of the analyses? If these were conducted, it might be worthwhile to indicate this briefly in the methods. 

Since the limitations mentioned are mostly methodological ones that cannot be corrected retrospectively, it is unfortunately not possible to include these in preliminary or other analyses and are thus described in the discussion as limitations of the current study

Results:

No additional suggestions here beyond the statement about the describing the model development in the methods section.

Discussion:

This section was well written and very interesting. 

Conclusions:

  1. Focus on the results from this study. Any similar results from other studies should be highlighted in the discussion section, in my opinion.

The Conclusions section has been rewritten in alignment with this comment.

Reviewer 2 Report

Zohar et al. hypothesized that maternal child feeding practices would mediate the effect of maternal authority style on child eating behaviors. They found that they were two distinct patterns of mother-child feeding and eating dynamics, apparently related to children with opposing appetitive tendencies. Overall, the study contains important information. The Results and Discussions were all well written.

Strength and limitation:

Strength: This study a cross-sectional study, embedded within a longitudinal study, that took place when the children were on average six years of age.

Limitation: Only Israeli Jews were enrolled.

Only a few editing errors and typos: line 10, 5.9+1.1, be +; to cite other’s work, please write Author et al.. Line 58, et al.; line 69, [22], no comma here; line 148, 0.84.; line 185, “+” meaning?; references 23, 41 and 42, some incompatible styles;

Author Response

Zohar et al. hypothesized that maternal child feeding practices would mediate the effect of maternal authority style on child eating behaviors. They found that they were two distinct patterns of mother-child feeding and eating dynamics, apparently related to children with opposing appetitive tendencies. Overall, the study contains important information. The Results and Discussions were all well written.

Strength and limitation:

Strength: This study a cross-sectional study, embedded within a longitudinal study, that took place when the children were on average six years of age.

Limitation: Only Israeli Jews were enrolled.

Only a few editing errors and typos:

  1. line 10, 5.9+1.1, be +;

This has been corrected.

  1. to cite other’s work, please write Author et al..

Line 58, et al.;

The work of Alfonso et al. [42]. Moreira et al. [49] and Mitchel et al. [48] is central to the current paper and different aspects of these three studies are described in different parts of the current paper including the Discussion.

  1. line 69, [22], no comma here;

This has been corrected.

  1. line 148, 0.84.;

This has been corrected.

  1. references 23, 41 and 42, some incompatible styles;

We could not spot a mistake in reference 23, but have made corrections to references no. 21, 22 and 42.

Reviewer 3 Report

The manuscript entitled “Two to tango? The Dance of Maternal Authority and Feeding Practices with Child Eating Behavior.” presents issues associated with relationship between maternal feeding practices and children’s eating problems.

Title:

  • Authors should formulate a more “scientific” title - formulated while using a proper scientific language, as their current title is rather formulated as for the column of the newspaper. The proper title should be rather informative than catchy. Such title suggest review type of article rather than research one.

Abstract:

  • Information about study protocol, methods applied should be presented.
  • Lines 10, 12, 22 – words “Methods”; “Results”; “Conclusion:” should be removed (structured abstracts, but without headings)
  • Please add the period when the study was conducted

Introduction:

  • Lines 79-80 – “A subset of children were followed for two 80 additional years and picky eating was measured [25]. This sentence is redundant. It brings nothing to the manuscript than self-citation.
  • In this section Authors presented the information associated with parenting styles/ feeding practices and child eating behaviors. This section should be briefly presented – what do we know and what is the background for this study. Some more detailed information about other studies are necessary (not just mentioned that such studies were conducted). The good background should present the history of problem, the current knowledge and scientific "gap", and then authors should present how their study could fill this gap to justify the study. Authors should emphasize the novelty of the study.

Materials and Methods:

  • Information about study protocol (e.g. recruitment, inclusion and exclusion criteria) .
  • Line 101 – Parental Authority Questionnaire - what is the original language of the questionnaire. Was the questionnaire translated? Who did so? Any validation of the translated questionnaire? Please specify it for PAQ, CEBQ, CFQ, OC.
  • Line 101 – “Parental Authority Questionnaire” it should be “Parental Authority Questionnaire (PAQ)”
  • Line 111 – “Child Eating Behavior Questionnaire (CEBQ, 30),” it should be “Child Eating Behavior Questionnaire (CEBQ) [30],”
  • Lines 120-122 – “A study conducted on 1002 low-income preschool-age children recruited from Head Start locations in the United States [31] showed good validity and good internal reliability for this scale (Cronbach α’s ≥ .70).” – this sentence is redundant in this place
  • Line 125 – “Child Feeding Questionnaire (CFQ, 10)” – it should be “Child Feeding Questionnaire (CFQ) [10]”
  • Lines 137-138 – “A Swedish study conducted on 876 138 mothers resulted in good test-retest reliability (between .56-.89) and good construct validity [32].” – this sentence is redundant in this place
  • Line 148 – “and 084. respectively" it should be “and 0.84 respectively”
  • More information should be presented for “Data Analysis” (including the references for cut-off for values of acceptance of the model – NFI, RMSEA, SRMR, etc.)

Results:

  • Line 159 – “Pearson correlations” - Was the normality of distribution tested? The information about it should be added and authors should be consequent. If data have normal distribution, they should be treated as such, if not, nonparametric tests should be applied. Please specify it.

Discussion:

  • This section should be improved. Authors should relate the findings to those of similar studies (please indicate the data) and point the differences and similarities between the studies. Please describe these studies more (the sample, data, etc.). What implications do your results have based on this research.
  • Line 229 – “have low BMIs” (How low? In normal range or underweight?) – please specify it
  • Line 243 – “on a similar group of women” – please specify it
  • Line 250 – “study of older children” – please specify it

Conclusion:

  • This section must be shortened. The conclusion must be more related to the findings (exclusively). Please remove the references (citation) in this section.

Minor comments:

  • Authors should follow the Instructions for authors while preparing their manuscript (e.g. reference style)

Author Response

The manuscript entitled “Two to tango? The Dance of Maternal Authority and Feeding Practices with Child Eating Behavior.” presents issues associated with relationship between maternal feeding practices and children’s eating problems.

Title:

  1. Authors should formulate a more “scientific” title - formulated while using a proper scientific language, as their current title is rather formulated as for the column of the newspaper. The proper title should be rather informative than catchy. Such title suggest review type of article rather than research one.

Although the title is metaphorical rather than literal we feel it captures the very core of the message of the current paper, and therefore, we have retained it.

Abstract:

  1. Information about study protocol, methods applied should be presented.

This information is now given in brief in the abstract and in more detail in the methods section, which now starts with a subsection “protocol”.

  1. Lines 10, 12, 22 – words “Methods”; “Results”; “Conclusion:” should be removed (structured abstracts, but without headings)

This has been corrected.

  1. Please add the period when the study was conducted

Now appears in the opening of the Methods chapter under protocol.

Introduction:

  1. Lines 79-80 – “A subset of children were followed for two 80 additional years and picky eating was measured [25]. This sentence is redundant. It brings nothing to the manuscript than self-citation.

This sentence has been moved for better flow of the text but not deleted. It is there to explain the wider frame of the study from which this cross-sectional snapshot has been extracted.

  1. In this section Authors presented the information associated with parenting styles/ feeding practices and child eating behaviors. This section should be briefly presented – what do we know and what is the background for this study.

Now presented in the second paragraph of the introduction.

  1. Some more detailed information about other studies are necessary (not just mentioned that such studies were conducted). The good background should present the history of problem, the current knowledge and scientific "gap", and then authors should present how their study could fill this gap to justify the study. Authors should emphasize the novelty of the study.

The scientific gap addressed by the current study has now been clearly addressed.

Materials and Methods:

  1. Information about study protocol (e.g. recruitment, inclusion and exclusion criteria).

This information has been added to the protocol paragraph of the Methods.

  1. Line 101 – Parental Authority Questionnaire - what is the original language of the \ questionnaire. Was the questionnaire translated? Who did so? Any validation of the translated questionnaire? Please specify it for PAQ, CEBQ, CFQ, OC.

The sources of the translations into Hebrew have been added for these scales.

  1. Line 101 – “Parental Authority Questionnaire” it should be “Parental Authority Questionnaire (PAQ)”

This has been corrected.

  1. Line 111 – “Child Eating Behavior Questionnaire (CEBQ, 30),” it should be “Child -Eating Behavior Questionnaire (CEBQ) [30],”

This has been corrected.

  1. Lines 120-122 – “A study conducted on 1002 low-income preschool-age children recruited from Head Start locations in the United States [31] showed good validity and good internal reliability for this scale (Cronbach α’s ≥ .70).” – this sentence is redundant in this place

This sentence has been omitted.

  1. Line 125 – “Child Feeding Questionnaire (CFQ, 10)” – it should be “Child Feeding Questionnaire (CFQ) [10]”

This has been corrected throughout the ms.

  1. Lines 137-138 – “A Swedish study conducted on 876 138 mothers resulted in good test-retest reliability (between .56-.89) and good construct validity [32].” – this sentence is redundant in this place

This sentence has been omitted.

  1. Line 148 – “and 084. respectively" it should be “and 0.84 respectively”

This has been corrected.

  1. More information should be presented for “Data Analysis” (including the references for cut-off for values of acceptance of the model – NFI, RMSEA, SRMR, etc.)

This has been added.

Results:

  1. Line 159 – “Pearson correlations” - Was the normality of distribution tested? The information about it should be added and authors should be consequent. If data have normal distribution, they should be treated as such, if not, nonparametric tests should be applied. Please specify it.

This has been added: All variables were tested for normality distribution and found to be adequate.

Discussion:

  1. This section should be improved. Authors should relate the findings to those of similar studies (please indicate the data) and point the differences and similarities between the studies. Please describe these studies more (the sample, data, etc.). What implications do your results have based on this research.

The Discussion has been expanded to better integrate the findings of the current study within the framework of related published results.

  1. Line 229 – “have low BMIs” (How low? In normal range or underweight?) – please specify it

This refers to a comparison between groups; the sentence has been worded more clearly: “Individuals retrospectively reporting higher maternal pressure to eat during their childhood tend to have lower BMIs and healthier eating attitudes [11] as adults than those reporting lower maternal pressure to eat during their childhood.”

  1. Line 243 – “on a similar group of women” – please specify it

This has been clarified.

  1. Line 250 – “study of older children” – please specify it

More information about the participants has been provided.

Conclusion:

  1. This section must be shortened. The conclusion must be more related to the findings (exclusively). Please remove the references (citation) in this section.

The Conclusions have been rewritten and references have been removed.

Minor comments:

  1. Authors should follow the Instructions for authors while preparing their manuscript (e.g. reference style)

According to the guidelines online, “references may be in any style, provided that you use the consistent formatting throughout”. We have revised the reference list so that the style is consistent throughout.

Round 2

Reviewer 3 Report

The authors corrected or explained most of my comments. However, I think that there is still room for improvement.

I am not convinced that that the presented title reflect the manuscript content but it I leave it to the Authors decision. Please note that this article may be omitted by search engines (e.g. in systematic reviews searches)

Line 176 – indicated that cutoff “root mean square error of approximation (RMSEA) < .08” is acceptable fit (not good fit). Could be better but it is ok.

The level of self-citation is still quite high – 6 references from 48 (12.5%)
